

# Identification of significant gene and pathways involved in HBV-related hepatocellular carcinoma by bioinformatics analysis

Shucai Xie[1], Xili Jiang[2], Jianquan Zhang[1], Shaowei Xie[1], Yongyong Hua[1], Rui Wang[1] and Yijun Yang[1]

[1] Department of Hepatobiliary Surgery, Haikou People's Hospital/Affiliated Haikou Hospital of Xiangya Medical College, Central South University, Haikou, Hainan, China
[2] Department of Radiology, The Second People's Hospital of Hunan Province/Brain Hospital of Hunan Province, Changsha, China

Corresponding author
Yijun Yang, dryyj@163.com

## ABSTRACT

**Background**. Hepatocellular carcinoma (HCC) is a common malignant tumor affecting the digestive system and causes serious financial burden worldwide. Hepatitis B virus (HBV) is the main causative agent of HCC in China. The present study aimed to investigate the potential mechanisms underlying HBV-related HCC and to identify core biomarkers by integrated bioinformatics analyses.

**Methods**. In the present study, HBV-related HCC GSE19665, GSE55092, GSE94660 and GSE121248 expression profiles were downloaded from the Gene Expression Omnibus database. These databases contain data for 299 samples, including 145 HBV-related HCC tissues and 154 non-cancerous tissues (from patients with chronic hepatitis B). The differentially expressed genes (DEGs) from each dataset were integrated and analyzed using the RobustRankAggreg (RRA) method and R software, and the integrated DEGs were identified. Subsequently, the gene ontology (GO) functional annotation and Kyoto Encyclopedia of Genes and Genomes (KEGG) pathway analysis were performed using the DAVID online tool, and the protein–protein interaction (PPI) network was constructed using STRING and visualized using Cytoscape software. Finally, hub genes were identified, and the cBioPortal online platform was used to analyze the association between the expression of hub genes and prognosis in HCC.

**Results**. First, 341 DEGs (117 upregulated and 224 downregulated) were identified from the four datasets. Next, GO analysis showed that the upregulated genes were mainly involved in cell cycle, mitotic spindle, and adenosine triphosphate binding. The majority of the downregulated genes were involved in oxidation reduction, extracellular region, and electron carrier activity. Signaling pathway analysis showed that the integrated DEGs shared common pathways in retinol metabolism, drug metabolism, tryptophan metabolism, caffeine metabolism, and metabolism of xenobiotics by cytochrome P450. The integrated DEG PPI network complex comprised 288 nodes, and two important modules with high degree were detected using the MCODE plug-in. The top ten hub genes identified from the PPI network were SHCBP1, FOXM1, KIF4A, ANLN, KIF15, KIF18A, FANCI, NEK2, ECT2, and RAD51AP1. Finally, survival analysis revealed that patients with HCC showing altered ANLN and KIF18A expression profiles showed
worse disease-free survival. Nonetheless, patients with FOXM1, NEK2, RAD51AP1, ANLN, and KIF18A alterations showed worse overall survival.

**Conclusions**. The present study identified key genes and pathways involved in HBV-related HCC, which improved our understanding of the mechanisms underlying the development and recurrence of HCC and identified candidate targets for the diagnosis and treatment of HBV-related HCC.

## INTRODUCTION

Hepatocellular carcinoma (HCC) is the sixth most commonly diagnosed type of malignant tumor and is the second leading cause of cancer-related deaths worldwide. It is estimated that there were about 841,000 new cases and 782,000 deaths caused by liver cancer worldwide in 2018 (*Bray et al., 2018*), with Chinese patients making up more than half of the global HCC burden (*Jemal et al., 2011*; *Bray et al., 2018*). The high incidence of HCC in parts of Asia is mainly due to the prevalence of hepatitis B virus and C virus infections, especially the hepatitis B virus (*Bray et al., 2018*). Accumulating evidence has shown that carcinogenesis and progression of HCC are closely related to overexpression of various oncogenes and inactivation of tumor suppressor genes.

The poor prognosis associated with HCC is attributed to the lack of effective diagnostic and therapeutic methods in the early stage of the disease. In recent years, gene targeting therapy has been increasingly used for the treatment of advanced HCC, and significant progress has been made. The most commonly reported genetic alterations in HCC include mutations in the TERT promoter, TP53, CTNB1, AXIN1, ARID1A, CDKN2A, ARID2, RPS6KA3, and CCND1 (*Khemlina, Ikeda & Kurzrock, 2017*). Sorafenib targets multiple kinases and has been approved by the US Food and Drug Administration for the treatment of advanced HCC (*Zhu et al., 2017*). However, sorafenib has many shortcomings, such as low efficiency, high cost, and multiple side effects (*Hu et al., 2013*). Therefore, there is an urgent need to explore the relationship between the new gene function and the occurrence, development, and malignant characteristics of HCC, as well as to elucidate the precise molecular mechanisms underlying HCC, develop early screening methods, and discover novel and effective therapeutic strategies.

Recently, high-throughput technologies and gene chips have served as rapid methods for the identification of differentially expressed genes (DEGs) and functional pathways involved in the initiation and development of various diseases (*Roh et al., 2010*; *Vogelstein et al., 2013*). In these studies, a large number of tumor samples can be analyzed and thousands of genes can be identified; as a result, bioinformatics methods have become necessary for the analysis of gene expression profiles. However, obtaining reliable results from a single gene expression profile data is difficult, considering the potentially large
number of differentially expressed genes, lack of stability and reproducibility, and high false-positive rates.

The RobustRankAggreg (RRA), which is based on a statistical model, is a biological analysis method for the integration and analysis of multiple gene lists (*Kolde et al., 2012*). RRA is a rank aggregation algorithm that assumes that all genes are arranged randomly in each dataset. A gene with a higher ranking in all datasets has a lower *P*-value and has a higher likelihood of being a DEG. Compared to other strategies used for the meta-analysis of datasets from multiple databases, the RRA method is more robust and easier to compute and facilitates better evaluation of the significance of the results. In addition, the RRA algorithm can handle the variable number of genes identified from different microarray platforms. More importantly, the RRA method does not strictly require the use of certain subset of problems or complete datasets to produce highly reliable results (*Kolde et al., 2012*). Therefore, the RRA algorithm is highly suitable for the integrated analysis of datasets from multiple databases.

In the present study, four GSE datasets GSE19665 (*Deng et al., 2010*), GSE55092 (*Melis et al., 2014*), GSE94660 (*Yoo et al., 2017*), and GSE121248 (*Wang, Ooi & Hui, 2007*) were downloaded from GEO; these datasets comprise a total of 299 samples, including 145 hepatitis B virus (HBV)-related HCC tissues and 154 non-cancerous tissues (chronic hepatitis B patients). The chip probe IDs were converted to their corresponding gene symbols. Bioinformatics analysis using R software and RRA method was then performed to obtain the integrated differentially expressed genes (DEGs). The Gene Ontology (GO; http://www.geneontology.org) is a public bioinformatics resource that provides information about gene product function using ontologies to represent biological knowledge (*Gaudet & Dessimoz, 2017*). KEGG (Kyoto Encyclopedia of Genes and Genomes) is a knowledgebase used for the systematic analysis of gene functions and for linking genomic information with higher-order functional information (*Kanehisa & Goto, 2000*). Herein, enriched GO terms and KEGG pathways were identified using the online tool DAVID 6.7. Then, the protein-protein interaction (PPI) network of the DEGs was constructed, and the hub genes were identified. We constructed the network using the hub genes and their co-expressed genes and analyzed the biological processes associated with the hub genes. Finally, survival analysis was performed based on the hub DEGs by generating the Kaplan–Meier curves in the cBioPortal. Therefore, the hub DEGs and the associated enriched pathways identified in this study can serve as reliable molecular markers for HBV-related HCC.

## MATERIAL AND METHODS

### Microarray data

Gene expression profiles of GSE19665, GSE55092, GSE94660 and GSE121248 were acquired from the National Center for Biotechnology Information (NCBI) Gene Expression Omnibus (GEO) database (https://www.ncbi.nlm.nih.gov/geo/). GSE19665, GSE55092 and GSE121248 dataset were based on the platforms of GPL570 [HG-U133_Plus_2] Affymetrix Human Genome U133 Plus 2.0 Array, while the platform of the GSE94660 dataset was
**Table 1** Details of the GEO HBV-related HCC data.

| GEO | Sample | Platform | Normal | Tumor | Reference |
|---|---|---|---|---|---|
| GSE19665 | hepatocellular carcinoma (HBV) | GPL570 | 5 | 5 | *Deng et al. (2010)* |
| GSE55092 | hepatocellular carcinoma (HBV) | GPL570 | 91 | 49 | *Melis et al. (2014)* |
| GSE94660 | hepatocellular carcinoma (HBV) | GPL16791 | 21 | 21 | *Yoo et al. (2017)* |
| GSE121248 | hepatocellular carcinoma (HBV) | GPL570 | 37 | 70 | *Wang, Ooi & Hui (2007)* |

**Notes.**

GEO, gene expression omnibus.

GPL16791 Illumina HiSeq 2500 (Homo sapiens). GSE19665, GSE55092, GSE94660, GSE121248 contain 5, 91, 21 and 37 cases of non-cancerous tissues from chronic hepatitis B patients, and 5, 49, 21 and 70 cases of HBV related HCC tissues respectively. The series matrix TXT files and platform TXT files of the four databases were downloaded separately, and the information is shown in Table 1. To obtain the international standard gene name, the process of the conversion of gene probe IDs in the matrix files to the gene symbols in the platform files was performed by using A Perl language command. Subsequently, the gene expression data, normalized by the normalization Between Arrays function, was subjected to log2 transformation in the limma R package (http://www.bioconductor.org/) (*Ritchie et al., 2015*). Mean values of $\log_2 FC$ was used when multiple probe sets are used for one gene.

## Screening for DEGs

The limma R package V3.5.2 in R software was used to identify DEGs in each dataset. The DEGs were screened out according to the cut-off criterion that adjusted *P*-value $< 0.05$ and $|\log_2 FC| > 1$. The RobustRankAggreg (RRA) R package (https://cran.rstudio.com/bin/windows/contrib/3.5/RobustRankAggreg_1.1.zip) (*Kolde et al., 2012*) was used to integrated and analyzed the four gene lists which were sorted by logFC value. The lists of significantly upregulated and downregulated genes were exported and saved as Excel files respectively.

## GO and KEGG pathway enrichment analyses of DEGs

The Database for Annotation, Visualization and Integrated Discovery (DAVID; http://david.ncifcrf.gov) (version 6.8) (*Dennis et al., 2003*; *Huang et al., 2007*), a common online program, integrates biological data and analysis tools to provide a comprehensive set of functional annotation information of large-scale lists of genes or proteins for users to grasp biological characteristics (*Huang, Sherman & Lempicki, 2009*). In order to understand the selected DEGs better, GO and KEGG pathway enrichment analysis were executed by using DAVID online tool. $P < 0.05$ was considered statistically significant.

## PPI network construction and module analysis

As a public online tool, the Search Tool for the Retrieval of Interacting Genes (STRING) (https://string-db.org/) is designed to construct a critical assessment and intergration of PPI network, including direct (physical) and indirect (function) association (*Szklarczyk et al., 2015*). To know the interactional correlation of the DEGs, PPI network was established by
**Table 2  Information of DEGs screened from each dataset.**

| GEO | Sample | Number of DEGs | Number of upregulated genes | Number of downregulated genes |
| --- | --- | --- | --- | --- |
| GSE19665 | hepatocellular carcinoma (HBV) | 648 | 257 | 391 |
| GSE55092 | hepatocellular carcinoma (HBV) | 1,034 | 409 | 634 |
| GSE94660 | hepatocellular carcinoma (HBV) | 1,171 | 360 | 811 |
| GSE121248 | hepatocellular carcinoma (HBV) | 580 | 167 | 413 |

Notes.
GEO, gene expression omnibus.

STRING and then displayed using Cytoscape software (3.7.1) that is a public bioinformatics software platform (*Shannon et al., 2003*). Furthermore, the plug-in Molecular Complex Detection (MCODE) app in Cytoscape software (*Bader & Hogue, 2003*) was also applied to select the significant modules of hub genes from the PPI network (degree cut-off $\geq 2$, node score cut-off $\geq 0.2$, K-core $\geq 2$, and max depth $= 100$). Moreover, the KEGG and GO analyses for DEGs in modules were used to investigate their potential information by using DAVID.

## Hub genes selection and analysis

The hub genes were selected with degrees $\geq 10$. The cBioPortal (http://www.cbioportal.org) online platform (*Cerami et al., 2012*; *Gao et al., 2013*) was used to analyze the network of the hub genes and their co-expression genes. The plug-in Biological Networks Gene Oncology tool (BiNGO) (version 3.0.3) (*Maere, Heymans & Kuiper, 2005*) in Cytoscape software was used to construct and visualize the biological process analysis of hub genes. The UCSC Cancer Genomics Browser (http://genome-cancer.ucsc.edu) (*Goldman et al., 2015*) was applied to construct hierarchical clustering of hub genes. The overall survival and disease-free survival of hub genes were analyzed using the Kaplan–Meier curve in the HCC datasets (TCGA, Provisional) of the cBioPortal. Comparison of expression of these genes in multiple databases were analyzed using online database Oncomine (http://www.oncomine.org) (*Rhodes et al., 2004*).

## RESULTS

### Identification of DEGs in HCC

Four HBV-related HCC gene expression profiles were downloaded from the NCBI GEO database. Afterwards, the gene expression data was normalized and DEGs were identified with the limma R package (adjusted $P < 0.05$ and |log fold change (FC)| $> 1$), and the results are shown in Fig. S1. We screened out 648, 1,043, 1,171, and 580 DEGs respectively (Table 2, Fig. 1, Table S1). Through the integration and analysis of RRA, a total of integrated 341 DEGs were identified from the four datasets, including 117 upregulated genes and 224 downregulated genes (Table S2). The top 20 upregulated genes and the top 20 downregulated genes were charted on a heat map, as shown in Fig. 2.

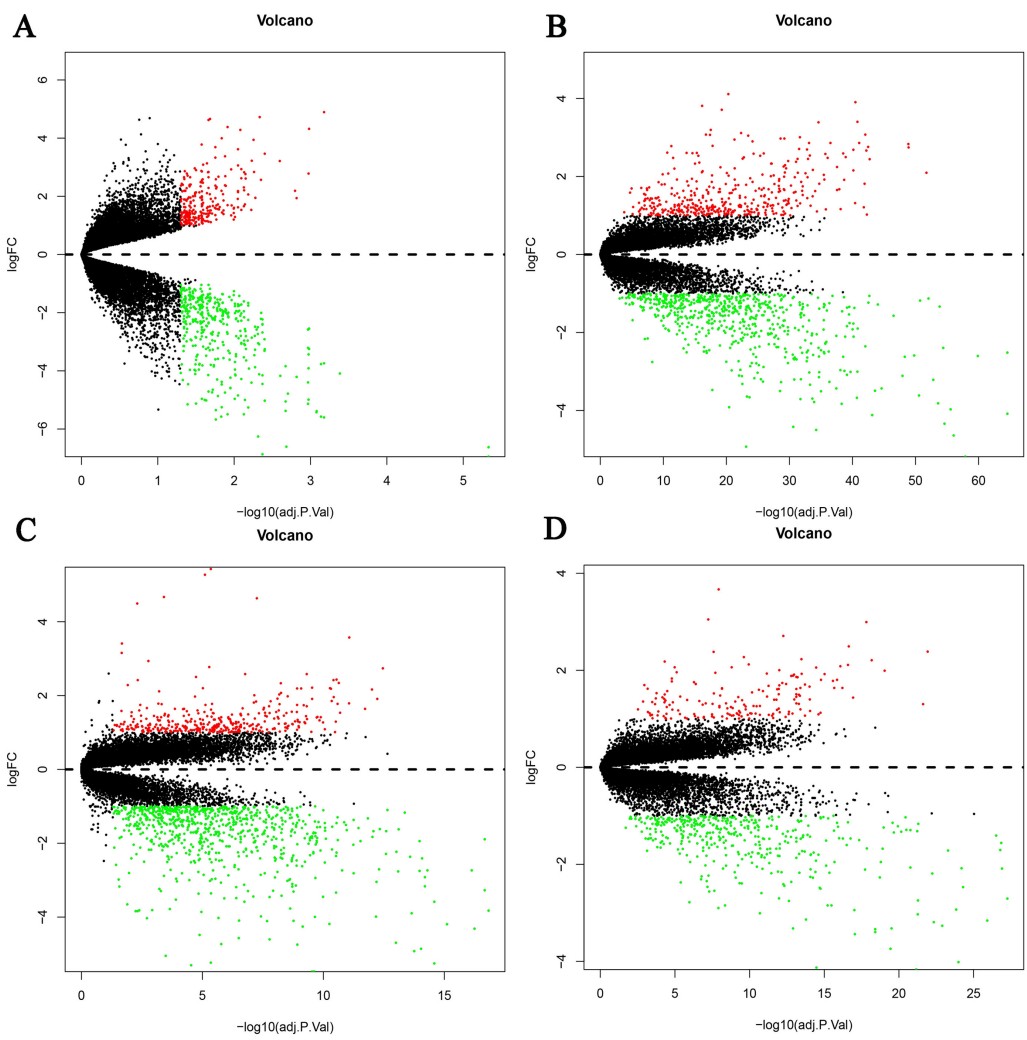

**Figure 1   Differential expression genes between the two groups of samples in each dataset.** (A) GSE19665, (B) GSE55092, (C) GSE94660, (D) GSE121248. The red dots represent the upregulated genes based on an adjusted *P* < 0.05 and log fold change > 1; the green dots represent the downregulated genes based on an adjusted *P* < 0.05 and log fold change < 1; the black spots represent genes with no significant difference in expression.

## Functional enrichment analyses of DEGs

To further investigate the biological functions of the 314 DEGs, GO analysis was performed using online database DAVID 6.7. As shown in Figs. 3A and 3B and Table 3, GO analysis can be divided into three functional groups: biological process group (BP), the cellular component group (CC), and the molecular function group (MF).The results of GO analysis exhibited that the integrated DEGs were particularly enriched in the BP, including cell cycle, M phase, cell cycle phase, mitosis and nuclear division for the upregulated DEGs and oxidation reduction, innate immune response, complement activation, response to wounding and activation of plasma proteins involved in acute inflammatory response for the downregulated DEGs. For the CC, the upregulated DEGs were mainly enriched

| GSE19665 | GSE55092 | GSE94660 | GSE121248 | Gene |
|---|---|---|---|---|
| 3.80 | 2.59 | 4.63 | 2.71 | GPC3 |
| 2.94 | 2.50 | 1.77 | 1.91 | FAM83D |
| 2.90 | 2.45 | 3.57 | 2.21 | TOP2A |
| 2.88 | 3.05 | 2.42 | 1.83 | LCN2 |
| 2.92 | 2.16 | 1.98 | 1.92 | ASPM |
| 2.84 | 2.83 | 2.01 | 2.39 | CAP2 |
| 2.65 | 2.54 | 1.53 | 2.09 | PRC1 |
| 2.62 | 2.68 | 2.42 | 1.89 | CCNB1 |
| 2.77 | 1.98 | 1.64 | 1.31 | IGF2BP3 |
| 1.29 | 4.12 | 2.94 | 3.67 | SPINK1 |
| 4.39 | 3.07 | 0.53 | 2.12 | RBM24 |
| 3.99 | 2.40 | 1.36 | 1.93 | NUF2 |
| 2.36 | 2.10 | 1.74 | 1.39 | KIF11 |
| 4.63 | 3.39 | 0.85 | 2.06 | ZIC2 |
| 2.48 | 2.25 | 1.30 | 1.93 | GINS1 |
| 3.77 | 3.08 | -0.05 | 2.18 | MAGEA6 |
| 2.66 | 2.58 | 1.27 | 2.23 | PBK |
| 2.23 | 3.40 | 1.84 | 2.00 | ROBO1 |
| 2.20 | 1.98 | 1.51 | 1.72 | KIF4A |
| 2.19 | 1.52 | 1.33 | 1.78 | NUSAP1 |
| -6.95 | -5.19 | -5.47 | -4.13 | HAMP |
| -5.60 | -3.97 | -4.86 | -3.32 | FCN3 |
| -5.33 | -3.69 | -4.10 | -2.85 | MT1M |
| -6.87 | -3.49 | -4.70 | -3.03 | CRHBP |
| -5.00 | -4.93 | -4.73 | -2.78 | C9 |
| -6.26 | -3.62 | -3.83 | -3.19 | CLEC4G |
| -4.71 | -3.50 | -3.99 | -3.74 | CNDP1 |
| -5.49 | -3.11 | -4.92 | -2.94 | TTC36 |
| -4.86 | -3.02 | -4.26 | -2.70 | APOF |
| -5.50 | -3.22 | -3.77 | -2.17 | SLC25A47 |
| -5.57 | -4.08 | -3.28 | -3.16 | CLEC1B |
| -4.46 | -2.79 | -3.78 | -2.23 | CYP2A6 |
| -4.15 | -3.15 | -5.48 | -3.40 | CYP1A2 |
| -5.25 | -4.42 | -3.10 | -2.54 | SLC22A1 |
| -4.07 | -2.62 | -3.42 | -2.10 | CD5L |
| -5.38 | -2.61 | -3.31 | -2.71 | CXCL14 |
| -4.23 | -2.52 | -4.32 | -2.08 | FCN2 |
| -4.04 | -2.86 | -2.87 | -2.31 | GYS2 |
| -4.99 | -4.34 | -2.77 | -3.27 | OIT3 |
| -3.83 | -2.36 | -4.61 | -2.34 | THRSP |

**Figure 2** **Log FC Heatmap of the top 20 DEGs (upregulated genes and downregulated genes) expression in all datasets.** The abscissa represent the GEO IDs, the ordinate represents the gene name, the red represents log FC > 0, the white represents log FC = 0, the green represents log FC < 0 and the value in the box represents the log FC value.

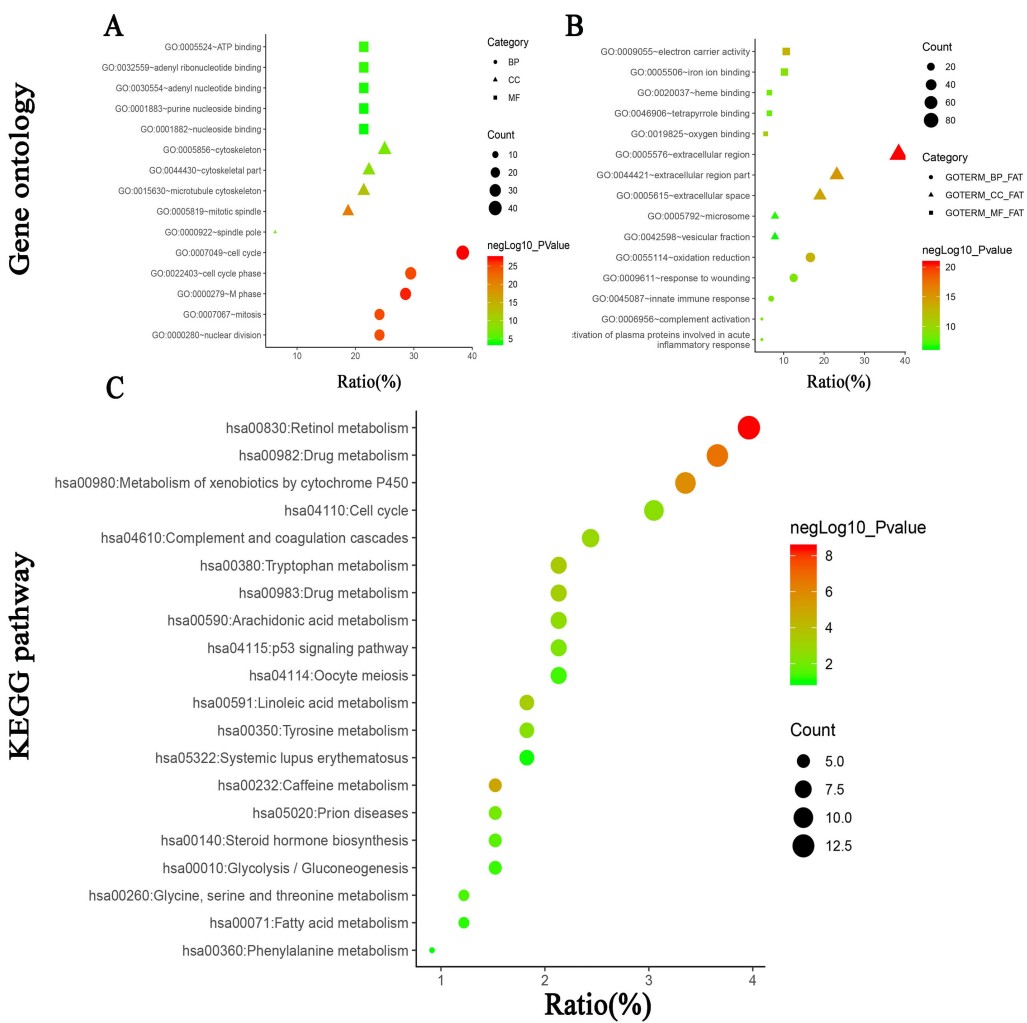

**Figure 3** (A) GO analysis of upregulated DEGs. (B) GO analysis of downregulated DEGs. (C) KEGG pathway of DEGs.

in mitotic spindle, microtubule cytoskeleton, cytoskeletal part, cytoskeleton and spindle pole and the downregulated DEGs were enriched in extracellular region, extracellular region part, extracellular space, microsome, vesicular fraction. In the MF, the upregulated DEGs were principally enriched in ATP binding, adenyl ribonucleotide binding, adenyl nucleotide binding, purine nucleoside binding, nucleoside binding, and the downregulated DEGs were enriched in electron carrier activity, oxygen binding, iron ion binding, heme binding, tetrapyrrole binding.

## Signaling pathway enrichment analysis

KEGG pathway enrichment analysis was performed using online database DAVID 6.7. As shown in Fig. 3C and Table 4, the results revealed that the integrated DEGs were particularly enriched in retinol metabolism, drug metabolism, metabolism of xenobiotics by cytochrome P450, caffeine metabolism and tryptophan metabolism.

**Table 3  Top 15 GO enrichment terms of differentially expressed genes associated with hepatitis B-related hepatocellular carcinoma.**

| Expression | Category | Term | Count | % | P value |
|---|---|---|---|---|---|
| upregulated | BP | GO:0007049~cell cycle | 43 | 38.39286 | 7.63E-28 |
| | BP | GO:0000279~M phase | 32 | 28.57143 | 2.48E-27 |
| | BP | GO:0022403~cell cycle phase | 33 | 29.46429 | 1.59E-25 |
| | BP | GO:0007067~mitosis | 27 | 24.10714 | 2.03E-25 |
| | BP | GO:0000280~nuclear division | 27 | 24.10714 | 2.03E-25 |
| | CC | GO:0005819~ mitotic spindle | 21 | 18.75 | 1.43E-21 |
| | CC | GO:0015630~microtubule cytoskeleton | 24 | 21.42857 | 2.84E-13 |
| | CC | GO:0044430~cytoskeletal part | 25 | 22.32143 | 3.19E-09 |
| | CC | GO:0005856~cytoskeleton | 28 | 25 | 5.91E-08 |
| | CC | GO:0000922~spindle pole | 7 | 6.25 | 6.75E-08 |
| | MF | GO:0005524~ATP binding | 24 | 21.42857 | 3.45E-05 |
| | MF | GO:0032559~adenyl ribonucleotide binding | 24 | 21.42857 | 4.27E-05 |
| | MF | GO:0030554~adenyl nucleotide binding | 24 | 21.42857 | 9.64E-05 |
| | MF | GO:0001883~purine nucleoside binding | 24 | 21.42857 | 1.22E-04 |
| | MF | GO:0001882~nucleoside binding | 24 | 21.42857 | 1.35E-04 |
| downregulated | BP | GO:0055114~oxidation reduction | 36 | 16.66667 | 7.03E-14 |
| | BP | GO:0045087~innate immune response | 15 | 6.944444 | 1.67E-09 |
| | BP | GO:0006956~complement activation | 10 | 4.62963 | 1.78E-09 |
| | BP | GO:0009611~response to wounding | 27 | 12.5 | 1.84E-09 |
| | BP | GO:0002541~activation of plasma proteins involved in acute inflammatory response | 10 | 4.62963 | 2.23E-09 |
| | CC | GO:0005576~extracellular region | 83 | 38.42593 | 2.05E-21 |
| | CC | GO:0044421~extracellular region part | 50 | 23.14815 | 9.40E-16 |
| | CC | GO:0005615~extracellular space | 41 | 18.98148 | 8.87E-15 |
| | CC | GO:0005792~microsome | 17 | 7.87037 | 2.50E-07 |
| | CC | GO:0042598~vesicular fraction | 17 | 7.87037 | 3.70E-07 |
| | MF | GO:0009055~electron carrier activity | 23 | 10.64815 | 1.17E-13 |
| | MF | GO:0019825~oxygen binding | 12 | 5.555556 | 5.61E-12 |
| | MF | GO:0005506~iron ion binding | 22 | 10.18519 | 5.88E-10 |
| | MF | GO:0020037~heme binding | 14 | 6.481481 | 6.05E-09 |
| | MF | GO:0046906~tetrapyrrole binding | 14 | 6.481481 | 1.33E-08 |

**Notes.**
BP, biological process; CC, cellular component; MF, molecular function; GO, gene ontology.

## Integration of PPI network and module analysis

PPI network of the 341 DEGs was established by STRING. A total of 288 DEGs (99 upregulated genes and 189 downregulated genes) were filtered into the DEG PPI network complex, which contained 288 nodes and 2,259 edges (Fig. S2, Tables S3 and S4). Among the 288 nodes, 59 central node genes were selected as hub genes with the criteria of filtering degree ≥ 10 (ie, each node has more than 10 connections/interactions). The top 10 hub genes were as follows: SHCBP1, FOXM1, KIF4A, ANLN, KIF15, KIF18A, FANCI, NEK2, ECT2 and RAD51AP1 (Table 5).
10.7717/peerj.7408
2019
Xie et al.

**Table 4** KEGG pathway analysis of differentially expressed genes associated with hepatitis B-related hepatocellular carcinoma.

| Term | Count | % | P value | Genes |
|---|---|---|---|---|
| hsa00830:Retinol metabolism | 13 | 3.963415 | 3.73E-09 | CYP3A4, CYP2B6, CYP2C9, CYP2C18, CYP2C8, CYP26A1, ADH1A, CYP1A2, CYP4A11, ADH4, CYP2A6, CYP2A7, RDH16 |
| hsa00982:Drug metabolism | 12 | 3.658537 | 2.06E-07 | CYP3A4, CYP2C18, CYP2C9, CYP2B6, ADH4, CYP2C8, GSTZ1, CYP2A6, ADH1A, CYP2A7, CYP1A2, ALDH3A1 |
| hsa00980:Metabolism of xenobiotics by cytochrome P450 | 11 | 3.353659 | 1.39E-06 | AKR1C3, CYP3A4, CYP2C18, CYP2C9, CYP2B6, ADH4, CYP2C8, GSTZ1, ADH1A, CYP1A2, ALDH3A1 |
| hsa00232:Caffeine metabolism | 5 | 1.52439 | 1.11E-05 | XDH, NAT2, CYP2A6, CYP2A7, CYP1A2 |
| hsa00380:Tryptophan metabolism | 7 | 2.134146 | 3.62E-04 | AADAT, TDO2, ACMSD, IDO2, KMO, CYP1A2, INMT |
| hsa00591:Linoleic acid metabolism | 6 | 1.829268 | 4.98E-04 | CYP3A4, CYP2C18, CYP2C9, AKR1B10, CYP2C8, CYP1A2 |
| hsa00983:Drug metabolism | 7 | 2.134146 | 5.42E-04 | CYP3A4, XDH, NAT2, CDA, CYP2A6, CYP2A7, TK1 |
| hsa04610:Complement and coagulation cascades | 8 | 2.439024 | 0.001332 | C8A, C8B, C7, C9, C6, KLKB1, F9, PLG |
| hsa00590:Arachidonic acid metabolism | 7 | 2.134146 | 0.002231 | AKR1C3, CYP4A11, CYP2C18, CYP2C9, CYP2B6, CYP2C8, CYP4F2 |
| hsa04110:Cell cycle | 10 | 3.04878 | 0.003227 | CCNE2, CCNB1, CDC6, MAD2L1, BUB1B, TTK, CDC20, MCM2, PTTG1, CCNA2 |
| hsa00350:Tyrosine metabolism | 6 | 1.829268 | 0.004034 | ADH4, GSTZ1, ADH1A, TAT, ALDH3A1, HPD |
| hsa04115:p53 signaling pathway | 7 | 2.134146 | 0.005932 | STEAP3, CCNE2, CCNB1, RRM2, IGF1, THBS1, IGFBP3 |
| hsa05020:Prion diseases | 5 | 1.52439 | 0.009832 | C8A, C8B, C7, C9, C6 |
| hsa00140:Steroid hormone biosynthesis | 5 | 1.52439 | 0.024978 | AKR1C3, CYP3A4, HSD17B2, SRD5A2, AKR1D1 |
| hsa00260:Glycine, serine and threonine metabolism | 4 | 1.219512 | 0.038725 | SDS, AGXT2, GNMT, GLDC |
| hsa04114:Oocyte meiosis | 7 | 2.134146 | 0.05109 | CCNE2, CCNB1, MAD2L1, IGF1, CDC20, AURKA, PTTG1 |
| hsa00010:Glycolysis/Gluconeogenesis | 5 | 1.52439 | 0.057701 | ADH4, ALDOB, FBP1, ADH1A, ALDH3A1 |
| hsa00071:Fatty acid metabolism | 4 | 1.219512 | 0.072811 | CYP4A11, ADH4, ADH1A, ACSL4 |
| hsa05322:Systemic lupus erythematosus | 6 | 1.829268 | 0.093168 | C8A, C8B, C7, C9, C6, HIST1H4H |
| hsa00360:Phenylalanine metabolism | 3 | 0.914634 | 0.099295 | TAT, ALDH3A1, HPD |

Furthermore, the two most significant modules (Figs. 4A and 4B, Tables S5 and S6) of the PPI network were selected for KEGG pathway enrichment analysis. Results showed that the genes in module 1 were mainly enriched in cell cycle, oocyte meiosis, p53 signaling pathway, progesterone-mediated oocyte maturation, and the genes in module 2 were mainly enriched in complement and coagulation cascades, prion diseases and systemic lupus erythematosus (Table 6).

## Hub gene selection and analysis

The biological process of the hub genes was analyzed and visualized using BiNGO in Cytoscape software and the result is shown in Fig. 5A. The network of hub genes and their co-expression genes was constructed using cBioPortal online platform. As show in Fig. 5B, the network contained 106 nodes, including 56 query genes and the 50 most frequently altered neighbor genes. Hierarchical cluster analysis showed that the high expression of hub genes was mainly in the region of HCC samples, whereas the low expression of hub genes

**Table 5  The top 10 most degree values hub genes between HBV-related HCC and normal samples.** Up indicated that the gene was identifed as up-regulated in HCC; Down indicated that the gene was reported as down-regulated. UN suggested the gene has not been reported in current HCC associated studies.

| Gene symbol | Gene description | logFC | Degree | Up/down |
|---|---|---|---|---|
| SHCBP1 | SHC binding and spindle associated 1 | 1.211943105 | 43 | Up |
| FOXM1 | forkhead box M1 | 1.832558689 | 43 | Up |
| KIF4A | kinesin family member 4A | 1.850277648 | 43 | Up |
| ANLN | anillin actin binding protein | 1.982343654 | 43 | Up |
| KIF15 | kinesin family member 15 | 1.247277869 | 43 | UN |
| KIF18A | kinesin family member 18A | 1.26437819 | 43 | Up |
| FANCI | FA complementation group I | 1.269987824 | 43 | UN |
| NEK2 | NIMA related kinase 2 | 1.652396787 | 43 | Down |
| ECT2 | epithelial cell transforming 2 | 1.418475417 | 43 | Up |
| RAD51AP1 | RAD51 associated protein 1 | 1.585151756 | 42 | UN |

**Notes.**

FC, fold change.

was mainly in the region of non-HCC samples (Fig. 4C). Subsequently, the prognostic information (overall survival and disease-free survival analyses) of the top 10 hub genes was available in the HCC datasets (TCGA, Provisional) of the cBioPortal online platform. HCC patients with ANLN and KIF18A alteration showed worse disease-free survival. Nonetheless, the patients with FOXM1, NEK2, RAD51AP1, ANLN, and KIF18A alteration showed worse overall survival (Fig. 6).

Among these genes, ANLN and KIF18A showed the highest node degrees with 43. Then, HCC patients with the two genes alteration showed worse overall survival and disease-free survival. Moreover, Oncomine analysis of cancer vs. normal tissue showed that ANLN and KIF18A were highly expressed in multiple HCC datasets (Fig. 7). These findings indicate that they may play important roles in the carcinogenesis or progression of HBV-HCC.

## DISCUSSION

HCC is a common malignant tumor affecting the digestive system. The incidence of HCC in developing countries is considerably higher than that in developed countries. Chronic infection with HBV or hepatitis C virus (HCV) is the primary etiological factor related to HCC in certain parts of Asia and sub-Saharan Africa (*Bray et al., 2018*). In western countries, the main risk factors include obesity, type 2 diabetes, heavy alcohol consumption, and smoking (*El-Serag, 2011*; *Mittal & El-Serag, 2013*). Although significant progress has been achieved in the diagnosis and treatment of HCC in recent years, the prognosis of HCC remains poor (*Okajima et al., 2017*). Thus, there is an urgent need to understand the detailed molecular mechanisms underlying HCC for early detection, diagnosis, and treatment of the disease. Recently, the wide application of microarray and high-throughput technologies has provided an effective way to screen thousands of genes

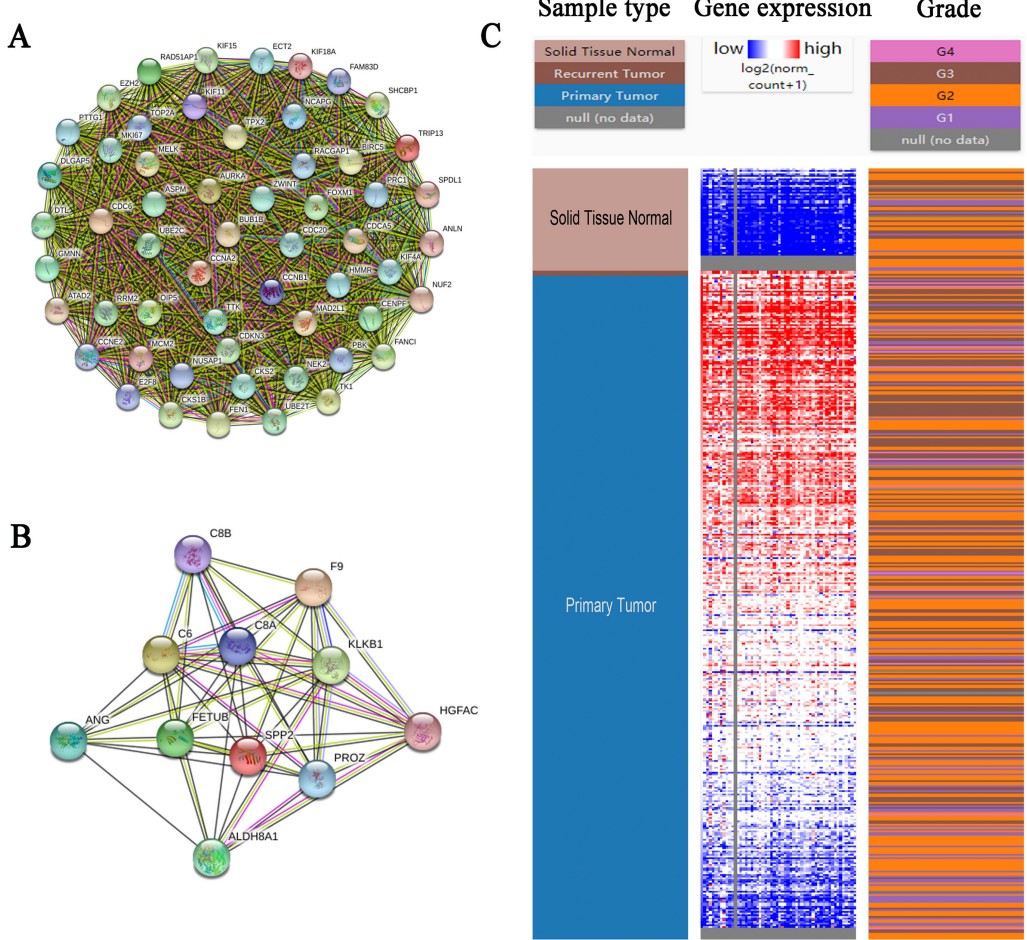

**Figure 4** **PPI network of module 1 (A), module 2 (B) and hierarchical clustering of hub genes was constructed using UCSC (C).** (A and B) Circles represent genes, lines represent interactions between gene-encoded proteins and line colors represent evidence of interactions between proteins. (C) The samples under the pink bar are normal samples and the samples under the blue bar are HCC samples. Upregulation of genes is marked in red; downregulation of genes is marked in blue.

that are likely to be involved in the occurrence and development of HCC. These genes could serve as potential targets for the diagnosis and treatment of HCC.

In the present study, we analyzed four mRNA expression datasets and identified a total of 341 DEGs, comprising 117 upregulated genes and 224 downregulated genes, in HBV-related HCC samples. GO and KEGG enrichment analyses showed that the upregulated genes were associated with the cell cycle (ontology: BP), mitotic spindle (ontology: CC), and adenosine triphosphate binding, adenyl ribonucleotide binding, adenyl nucleotide binding, purine nucleoside binding, and nucleoside binding (ontology: MF). The majority of the downregulated genes were enriched in oxidation reduction (ontology: BP), extracellular region (ontology: CC), and electron carrier activity (ontology: MF). The above results suggested that these DEGs are involved in the proliferation and cell cycle of chronic hepatitis B-induced HCC cells. KEGG pathway analysis revealed that

**Table 6  KEGG enrichment of genes in the top 2 modules.**

| Module | Term | Count | % | P value | Genes |
|--------|------|-------|---|---------|-------|
| Modul1 | hsa04110:Cell cycle | 10 | 18.18182 | 5.06E-11 | CCNE2, CCNB1, CDC6, MAD2L1, BUB1B, TTK, CDC20, MCM2, PTTG1, CCNA2 |
| | hsa04114:Oocyte meiosis | 6 | 10.90909 | 2.18E-05 | CCNE2, CCNB1, MAD2L1, CDC20, AURKA, PTTG1 |
| | hsa04115:p53 signaling pathway | 3 | 5.454545 | 0.021056 | CCNE2, CCNB1, RRM2 |
| | hsa04914:Progesterone-mediated oocyte maturation | 3 | 5.454545 | 0.032616 | CCNB1, MAD2L1, CCNA2 |
| Modul2 | hsa04610:Complement and coagulation cascades | 5 | 45.45455 | 3.11E-08 | C8A, C8B, C6, KLKB1, F9 |
| | hsa05020:Prion diseases | 3 | 27.27273 | 2.74E-04 | C8A, C8B, C6 |
| | hsa05322:Systemic lupus erythematosus | 3 | 27.27273 | 0.002195 | C8A, C8B, C6 |

the integrated DEGs were significantly enriched in retinol metabolism, drug metabolism, metabolism of xenobiotics by cytochrome P450, caffeine metabolism, and tryptophan metabolism. Previous studies have shown that dysregulation of the cell cycle and oxidation reduction play vital roles in the initiation or progression of tumors (*Choi et al., 2001*; *Wang et al., 2017*). Cytochrome P450 enzymes are involved in various types of cancer via several mechanisms, including the catalysis of the bioactivation of chemical procarcinogens, participation in the activation of cancer therapeutic drugs, as targets for cancer treatment, and as metabolic enzymes (*Hrycay & Bandiera, 2015*).

After screening out two modules with the most significant interactions, ten genes with the highest degrees of interaction were subsequently identified by constructing the PPI network. Results of survival analysis showed that the patients with alterations in FOXM1, NEK2, RAD51AP1, ANLN, and KIF18A expression patterns showed worse prognosis.

ANLN, an actin-binding protein, has been reported to be dysregulated in diverse human tumors (*Hall et al., 2005*). ANLN is not only considered as a proliferative marker, but also a prognostic and therapeutic indicator. ANLN knockdown was demonstrated to inhibit cell growth and migration in breast cancer (*Zhou et al., 2015*). In addition, another study showed that high ANLN levels in the nuclear fraction are involved in the cell cycle and are correlated with poor prognosis (*Magnusson et al., 2016*). Upregulation of ANLN expression plays an important role in non-small cell lung cancers (NSCLC) by activating RHOA and participating in the phosphoinositide 3-kinase/AKT pathway (*Suzuki et al., 2005*). A previous study showed that ANLN expression in gastric cancer (GC) is a molecular predictor of intestinal and proliferative type gastric tumors (*Pandi et al., 2014*). Furthermore, ANLN was identified as a biomarker for the prognosis of bladder urothelial carcinoma (*Zeng et al., 2017*), colorectal cancer (*Wang et al., 2016*), and lung adenocarcinoma (*Long et al., 2018*).

ANLN mRNA levels were upregulated in human HCC tissues compared to non-tumor liver tissues. ANLN knockdown was demonstrated to slow down the growth rates of tumors, reduce the number of tumors, and prolong the survival of mice (*Zhang et al., 2018*). Consistent with the findings reported in previous studies (*Lian et al., 2018*), our results showed that higher ANLN expression levels are associated with worse clinical
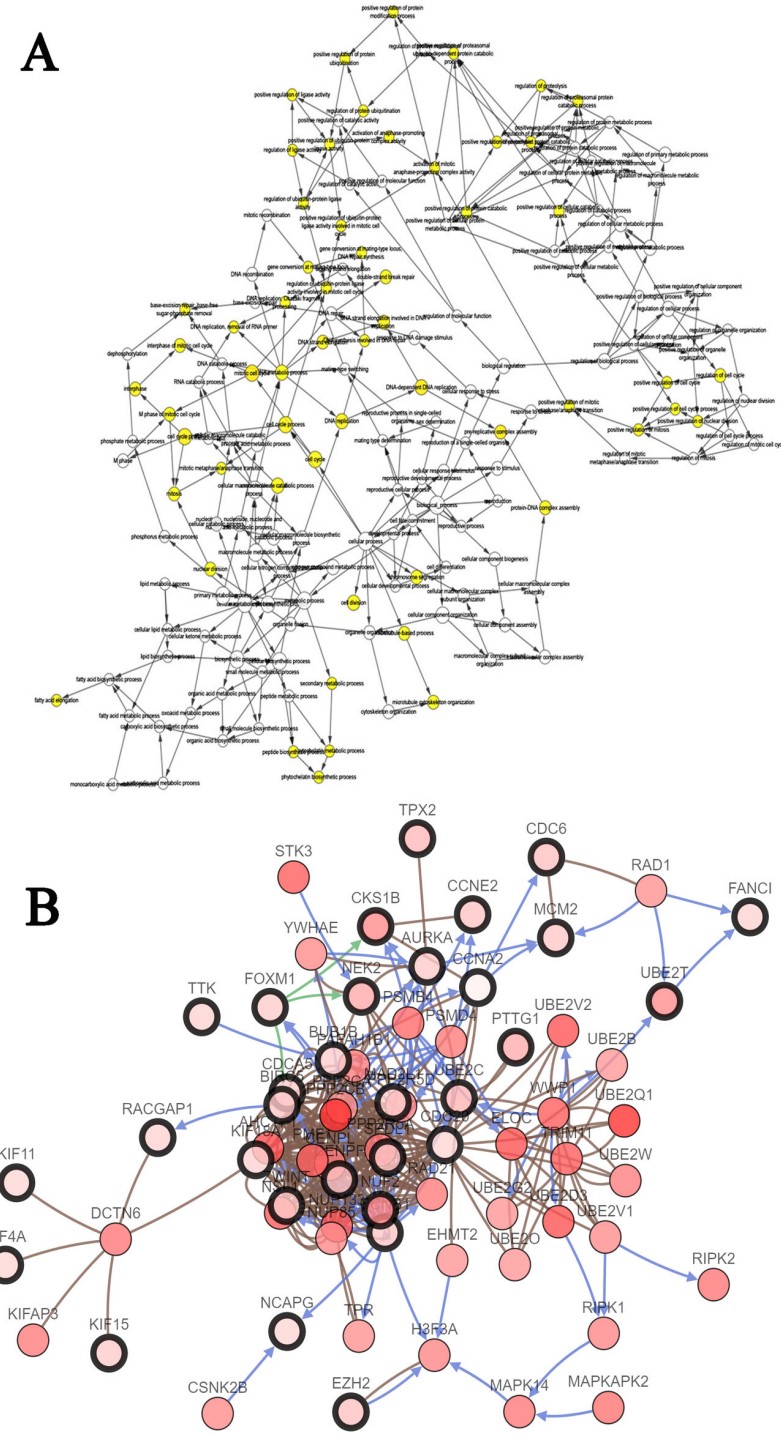

**Figure 5** **The biological process analysis of the hub genes.** (A) The biological process analysis of hub genes was constructed using BiNGO. The color depth of nodes refers to the corrected *P*-value of ontologies. The size of nodes refers to the numbers of genes that are involved in the ontologies. $P < 0.05$ was considered statistically significant. (B) Hub genes and their co-expression genes were analyzed using cBioPortal. Nodes with bold black outline represent hub genes. Nodes with thin black outline represent the co-expression genes.

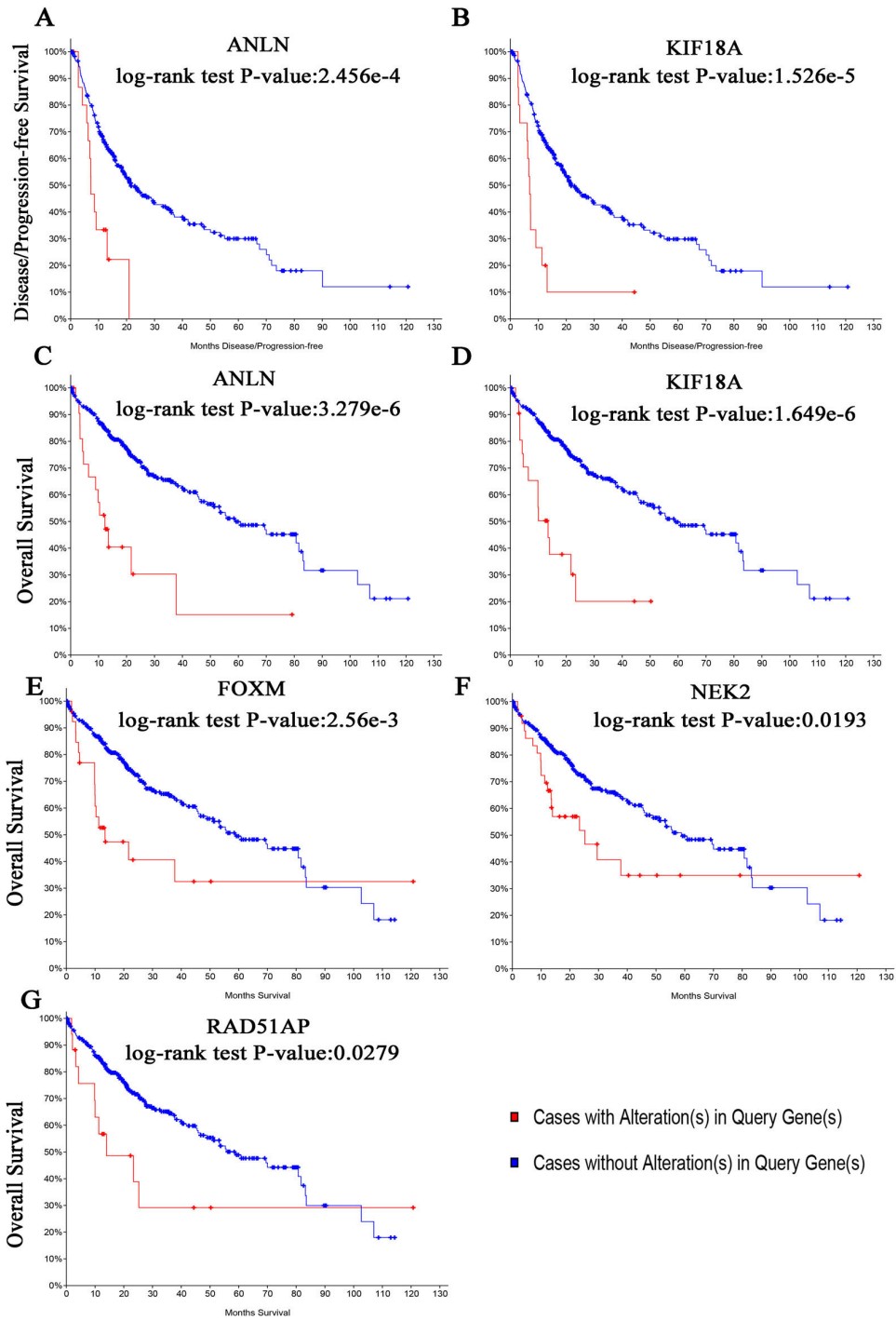

**Figure 6** (A, B) Disease-free survival analyses and (C–G) overall survival of hub genes were performed using cBioPortal online platform. $P < 0.05$ was considered statistically significant.

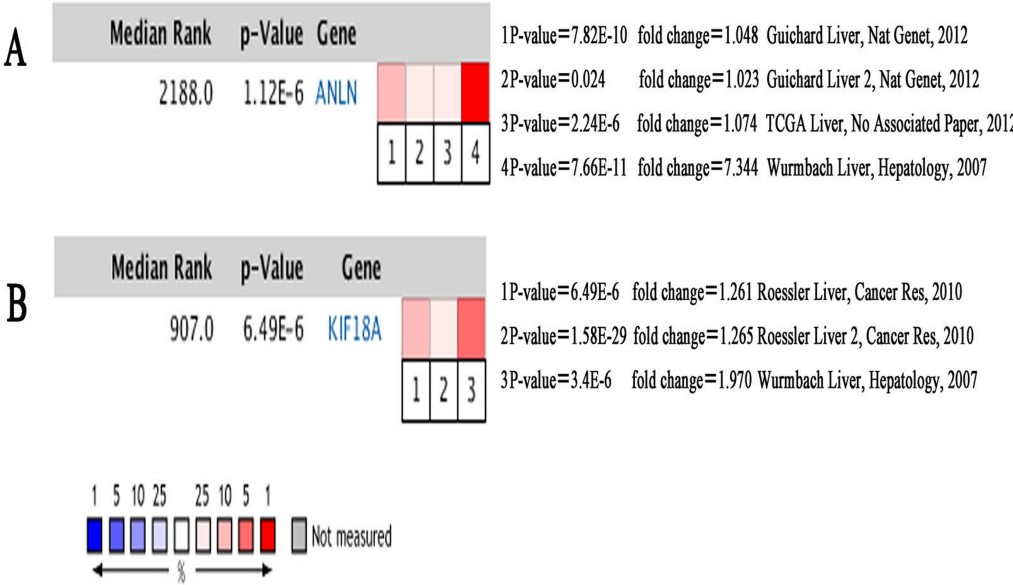

**Figure 7** Heat maps of ANLN (A) and KIF18A (B) gene expression in multiple clinical hepatocellular carcinoma samples vs. normal tissues using Oncomine analysis.

outcomes and a shorter survival times of patients with HCC, thereby highlighting the potential use of ANLN as a prognostic biomarker.

As a member of the kinesin-8 family, KIF18A plays crucial roles in regulating microtubule dynamics, chromosome congression, and cell division (*Mayr et al., 2007*). In fact, elevated KIF18A expression was observed in several human cancers. KIF18A overexpression in human breast cancer has been closely associated with tumor grade, metastasis, and poor survival (*Zhang et al., 2010*; *Kasahara et al., 2016*). Some researchers even advocate measurement of KIF18A levels in patients with estrogen receptor positive (ER+) breast cancer (BC) prior to receiving endocrine therapy (*Alfarsi et al., 2019*). KIF18A expression levels were found to positively contribute to tumor stage, lymphatic invasion, lymph node metastasis venous invasion, and peritoneal dissemination in CRC (*Nagahara et al., 2011*). Proteomic analysis indicated that KIF18A is a promising biomarker for the early diagnosis of cholangiocarcinoma (CCA) (*Rucksaken et al., 2012*).

KIF18A expression levels were found to be markedly higher in liver cancer tissues compared to adjacent normal liver tissues (*Liao et al., 2014*). KIF18A has been suggested to promote proliferation, invasion, and metastasis of HCC cells by activating cell cycle signaling pathway and the Akt and MMP-7/MMP-9-related signaling pathways (*Luo et al., 2018*). KIF18A levels were found to be significantly related to clinicopathologic factors associated with alpha-fetoprotein (AFP) concentrations ($\geq$200 ng/mL), tumor size ($\geq$5 cm), clinical tumor-node-metastasis (TNM) stage, and portal vein tumor thrombus (PVTT). In survival analysis, TCGA, Provisional higher KIF18A expression had worse prognosis (shorter DFS and OS) (*Liao et al., 2014*). The above results indicated that KIF18A could serve as a novel biomarker for the diagnosis and treatment of HCC. Our

findings are consistent with previous studies and demonstrated that previous experiments did not show whether these HCC were associated with HBV. Therefore, the role of KIF18A in HBV-related HCC types should be verified by further experiments.

As a member of the forkhead box (Fox) transcription factor family, FOXM1 is acts as an oncogene in many tumors, such as breast, cervix, and prostate cancers, and is known to play crucial roles in the prognosis and chemoresistance of tumors (*Zhu et al., 2018*). FOXM1 mRNA levels were upregulated in human HCC tissues and had positive relevance to the development, metastasis, recurrence, and worse clinical outcomes in HCC patients after orthotopic liver transplantation (*Sun et al., 2011*; *Dai et al., 2015*). SHCBP1, KIF4A, and ECT2 have been reported to mediate tumor initiation and progression of human HCC (*Tao et al., 2013*; *Chen et al., 2015*; *Hou et al., 2017*). NEK2 could serve as a useful predictor and potential therapeutic target in HCC (*Fu et al., 2017*). However, previous studies reported low NEK2 expression levels, which are inconsistent with our current findings. Other research groups reported that abnormal KIF15 levels were evidently associated with HCC progression and prognosis (*Chen et al., 2017*); however, these findings were not verified by cell or animal experiments. Similarly, these studies did not show that the occurrence of these HCCs is closely related to HBV.

FANCI and RAD51AP1 have been identified as new markers for HBV-related HCC, but have not been widely reported based on literature retrieval. Some findings provided new insights that RAD51AP1 is likely to mediate the molecular mechanisms underlying HCV-induced pathogenesis (*Nguyen et al., 2018*). However, further studies are required to verify the exact roles of these two genes.

In accordance with our findings, previous studies have also identified DEGs that participate in HBV-related liver cancer (*Zhou et al., 2017*). For example, Zhou et al. analyzed the gene expression profiles of GSE14520 and HCC samples from the Zhongshan Hospital affiliated with Fudan University, which comprised 63 paired HCC and non-tumor samples. All patients of these two cohorts were infected with hepatitis B virus. A total of 965 DEGs (389 upregulated genes and 576 downregulated genes) that were differentially regulated by at least two-fold with statistical significance. HSP90AB1, RPL8, NPM1, and MCM3 were selected as the hub genes from the PPI network. Nevertheless, the study by Zhou et al. comprised relatively fewer samples (66 primary HCC tumors and paired adjacent non-tumor tissues), and the main purpose of the study was to identify copy number variation (CNV)-driven DEGs. In another study, the DEGs that are common from four datasets were visualized using a Venn diagram (*Chen et al., 2019*). As a result, the number of DEGs obtained using this method was relatively small (84 upregulated and 46 downregulated). In the end, the following top ten hub genes were obtained: TOP2A, RFC4, CCNB1, CDC20, CDKN3, BUB1B, CCNB2, TPX2, PEN1, and MAD2L1. Compared to the previous two studies, we analyzed four GEO datasets comprising 299 samples, and 341 DEGs (117 upregulated and 224 downregulated) were identified. In addition, data analysis was conducted using the RobustRankAggreg (RRA) method, which is highly suitable for the analysis of datasets from multiple databases. Therefore, the hub genes identified in the present study are more reliable and comprehensive.

## CONCLUSION

In summary, by conducting an integrated bioinformatics analysis using multiple datasets, we identified DEGs and the association pathways involved in HBV-induced HCCs. In addition, we identified several key candidate genes and biological pathways that can provide a deeper and more comprehensive understanding of the occurrence and development of HCC and its association with HBV. Our findings provided valuable insights for the identification of novel biomarkers for the diagnosis and treatment of HCC.

### Funding

This work was supported by the Key Project of Science & Technology Development Fund of Hainan Province (No. ZDYF2018133&ZDXM2015082, ZDYF2019147). The funders had no role in study design, data collection and analysis, decision to publish, or preparation of the manuscript.

### Grant Disclosures

The following grant information was disclosed by the authors:
Key Project of Science & Technology Development Fund of Hainan Province: ZDYF2018133 & ZDXM2015082, ZDYF2019147.

### Competing Interests

The authors declare there are no competing interests.

### Author Contributions

- Shucai Xie conceived and designed the experiments, performed the experiments, analyzed the data, contributed reagents/materials/analysis tools, prepared figures and/or tables, authored or reviewed drafts of the paper, approved the final draft.
- Xili Jiang conceived and designed the experiments, performed the experiments, authored or reviewed drafts of the paper.
- Jianquan Zhang and Yijun Yang conceived and designed the experiments.
- Shaowei Xie performed the experiments, contributed reagents/materials/analysis tools, prepared figures and/or tables.
- Yongyong Hua contributed reagents/materials/analysis tools, prepared figures and/or tables.
- Rui Wang prepared figures and/or tables, authored or reviewed drafts of the paper.

### Data Availability

The following information was supplied regarding data availability: The raw measurements are available in the Supplemental Files.

### Supplemental Information

Supplemental information for this article can be found online at http://dx.doi.org/10.7717/peerj.7408#supplemental-information.

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
