# Peer review of "Identification of significant gene and pathways involved in HBV-related hepatocellular carcinoma by bioinformatics analysis"

_PeerJ, doi:10.7717/peerj.7408_

## Round 0.1 · original submission · Major Revisions

It is important that in the revised version of the manuscript you address the concerns raised by the reviewers, especially regarding

- the statistical significance and robustness of the results (using and reporting statistical tests correctly, justification of cutoff values)
- reproducibility of the findings (detailed description of methods, referencing, versions)
- and comparisons with other relevant studies.

·

Basic reporting

Xie et al. used gene expression data from diseased and healthy tissues to identify differentially expressed genes, aggregated using a rank-based method. They went on to perform enrichment analyses in terms of GO processes and KEGG pathways. They then created a PPI network from the DEGs and coexpressed genes and used a Cytoscape plugin to identify important genetic drivers (FOXM1, NEK2, RAD51AP1, ANLN and KIF18A) for the prognosis of HBV-related hepatocellular carcinoma, with survival analysis serving as validation for their findings. The findings are interesting and useful for further research into the diagnosis and treatment of the disease. The work sound but it has multiple problems which need to be addressed before publication.

The paper has poor English which make it hard to understand at times. For an example of spelling, on line 72 “tumor simples” should be “tumor samples”. In terms of grammar, lines 75 and 76 are not correct “difficult to obtain […] because of too many genes were identified”. There are cases such as line 323 where an abbreviation is used without prior definition, “CNV-driven”. I recommend that a native English speaker should read the manuscript and correct the mistakes to make it easier to understand. In the results section the cellular compartment “spindle” is quite meaningless if not mentioned the term fully (mitotic spindle astral microtubule end) or at least mitotic spindle.

The authors must take care to reference thoroughly, including every tool or software they use, and citing the paper and not just the web-link. For example, line 135 mentions STRING without citing the paper, as well as 124 with DAVID. In line 116 the authors mention using the limma R package and do not make any citations. Same with the used annotation databases (KEGG, GO). Also please cite R itself as well.

Figures and tables are generally good, however, I suggest to move some figures to supplementary and delete some altogether.

Move Figure 1 to supplementary.

Figure 2 is a useful figure, a minor point would be that volcano plots are conventionally plotted with logFC on the x-axis and -log10(adj.P.val) on the y-axis.

In Figure 3 the colour key must be corrected to show that 0 corresponds to white.

Figure 5A is not clear, I recommend that it be removed as it does not contribute to an increased understanding. Figure 5C has a mistake in the header “Sinple type”.

Experimental design

During the analysis, the authors made a mistake using gene names to map the probe sets. It is a dangerous practice because gene names are not unique. I strongly suggest redoing the analysis with mapping to any IDs which is in all the chips annotation files.

The methods lack the mention of how the authors handled when multiple probe sets are used for one gene. Please state that in the manuscript.

In line 130 they mention using P < 0.05 as a significance cut-off but do not specify whether this p-value is adjusted for multiple testing. Unclear whether this is a methodology issue or a typo.

The authors should describe why do they have chosen certain methods and cut-offs. For example (line 77) they use a rank aggregation algorithm but do not mention why they used this method and not others. A basic summary of other related methods and why the RRA method is the best choice, in this case, would greatly benefit the paper. This also applies to the choice of pathway database (KEGG and GO), why these and not e.g. Reactome or WikiPathway?

The authors mention various cut-offs, such as in line 138 when selecting significant modules, but do not justify why they chose these cut-offs. Were other parameters tried, with the final selection being the optimal choice? It is not clear in this case.

The authors used STRING database for PPIs but they not mentioned whether they used any filters regarding STRING. STRING collects the data by text-mining which made it inherent for false positive predictions. Have the authors considered this?

Validity of the findings

The findings, especially with the validation from an outside database, are extremely good and are an important addition to determining HCC outcome. However, these findings are depended on the used cut-offs, methods and gene mapping. The authors should redo the analysis or justify their cut-offs.

Additional comments

It is valuable work but needs some adjustments.

The main concerns raised are:
1. Incorrect mapping from probesets to genes - it can make the findings different
2. Methods not being described or justified sufficiently
3. Poor use of English language and grammar - makes the paper difficult to read
4. Insufficient or incorrect referencing
5. Some figures being unclear or not needed

Please address these before publication.

Reviewer 2 ·

Basic reporting

Article figures (especially Figure 5 and 6) are low resolution (at least the versions I had access to). Large part of the network in Supplementary Figure is missing.

Experimental design

The authors perform meta-analysis of the 4 different gene expression studies using Robust RankAggreg (RRA) method. I found several problems with data preprocessing and also some missing details of methods.

The used gene expression studies are referred as microarray experiments, however one of them (GSE94660) is RNAseq (which has no mayor consequence in the meta-analysis, but should be corrected). Also the authors write (line 113) that "gene expression were subjected to log2 transformation". Based on Fig. 1 it looks like that same used data (Fig.1 C,E and G?) were already log transformed, so this should be checked (as a second log transformation can influence the filtering criteria for differentially expressed genes (DEG)).

Identification of DEGs can be influenced by different non-biological effects (like batch effects), so some control for this in the individual datasets would be important to see (like a PCA plot of different samples coloured by tumor / control etc.).

In Figure 8 the authors performed survival analysis on cBioPortal. However the information about the used cancer dataset (TCGA?) and genomic profile (mutation, CNV alteration of gene expression alteration) is missing from the manuscript. If they used gene expression alteration, it would be important to see whether the decreased survival is associated with over or under-expression of the selected genes.

Validity of the findings

In Figure 9 the authors show association of ANLN and KIF18A with different tumor phenotypes (as Grade and Vascular Invasion). However I was not able to find the significance of this difference in the text or in the Figure. It would be also interesting to see this analysis with the other selected genes.

As the authors mention, there are other studies investigating HBV-related HCC gene expression (like reference Zhou et al. 2017, or Zhou et al. 2014 Int J Clin Exp Med ) Some more detailed comparison with these other studies (overlap of identified DEGs of functions) would be important information.

The authors performed KEGG pathway analysis with the DEGs. Comparing the enrichment of KEGG pathways with over and under expressed genes (like for GO terms) would be also an important information.

Additional comments

The manuscripts describes a meta-analysis of 4 different HBV-related hepatocellular carcinoma (HCC) gene expression studies and performs functional analysis of the identified differentially expressed genes (DSGs). However as they compare only HBV-related HCC samples against control (also from HBV infected patients) they analysis does not contain specific information about the HBV-related HCC (compared to other HCCs like HCV related or toxic) but HCC generally. This should be more emphasised in the manuscript.

---

## Round 0.2 · Major Revisions

The reviewer still found some major problems in the manuscript. Please try to address these questions in the revision

Reviewer 2 ·

Basic reporting

my concerns have been answered

Experimental design

1) Original question: In Figure 8 the authors performed survival analysis on cBioPortal. However the information about the used cancer dataset (TCGA?) and genomic profile (mutation, CNV alteration of gene expression alteration) is missing from the manuscript. If they used gene expression alteration, it would be important to see whether the decreased survival is associated with over or under-expression of the selected genes.

Response:Our research is to identify differentially expressed genes. The oncomine database was used to performed survival analysis of these genes. If we want to study the mutation and CNV alteration of gene expression alteration of gene, more work needs to be done.

Question: The authors say in their answer that they used “The oncomine database” for survival analysis. However based on Methods, they refer cBioPortal for survival analysis (line 146), and Oncomine for “interactional correlation of the expression level of hub genes and tumor grades, satellites, and 148  vascular invasion” (line 148). Which one is correct? If they used cBioProtal, I still did not get answer which dataset they used it. In cBioProtal there are 7 different HCC datasets (from MSK, INSERM, MSK, AMC, RIKEN, and 2 from TCGA), so it is an important information for the reader (and for the reproducibility of the analysis) which dataset they used. Also they say in results that “HBV-related liver cancer with ANLN and KIF18A alteration showed worse disease-free survival“. As far as I know non of these dataset is specific to HBV related cancer (but generally HCC), so the term HBV-related liver cancer should be used with caution here.

Validity of the findings

2) Original question: In Figure 9 the authors show association of ANLN and KIF18A with different tumor phenotypes (as Grade and Vascular Invasion). However I was not able to find the significance of this difference in the text or in the Figure. It would be also interesting to see this analysis with the other selected genes.

Response: Figure 9 (now is Figure 8) is from Oncomine. Although there are no results of statistical analysis, we can still find that the expression of these two genes is correlated with the phenotype of tumor. In survival analysis, only these two genes (ANLN and KIF18A) alteration showed worse disease-free survival and worse overall survival. Secondly, these two genes are the focus of this analysis. So, we want to study the relationship between their expression and phenotype of tumor. When we searched for the correlation between the expression of other genes and the phenotype of tumors in the same dataset (Wurmbach Liver dataset), some of them lacked statistical data, while others did not show a significant correlation with the phenotype of tumors (the results were not shown).

Question: I still think that if the authors write “In the Wurmbach Liver dataset, higher expression of ANLN and KIF18A had significantly correlated with tumor grade, satellites, and vascular invasion” (line 220), than they should use some statistical analysis to support their conclusions.

3) Original question: The authors performed KEGG pathway analysis with the DEGs. Comparing the enrichment of KEGG pathways with over and under expressed genes (like for GO terms) would be also an important information.

Response: Previous studies have confirmed that comparing the enrichment of KEGG pathways with over and under expressed genes (like for GO terms) will lead to a reduction in the number of meaningful signaling pathways.

Question: Could the authors cite some relevant literature for this claim?

Additional comments

The authors have answered some of my questions, but at other points the clear answers are still missing.

---

## Round 0.3 · accepted · Accept

All the previous comments have been addressed

Reviewer 2 ·

Basic reporting

The authors have answered my questions.

Experimental design

The authors have answered my questions.

Validity of the findings

The authors have answered my questions.